# Combined Mutational and Clonality Analyses Support the Existence of Intra-Tumor Heterogeneity in Papillary Thyroid Cancer

**DOI:** 10.3390/jcm10122645

**Published:** 2021-06-16

**Authors:** Marina Muzza, Gabriele Pogliaghi, Luca Persani, Laura Fugazzola, Carla Colombo

**Affiliations:** 1Laboratory of Endocrine and Metabolic Research, Istituto Auxologico Italiano IRCCS, 20095 Milan, Italy; marina.muzza@unimi.it (M.M.); gabriele.pogliaghi@unimi.it (G.P.); 2Division of Endocrine and Metabolic Diseases, Istituto Auxologico Italiano IRCCS, 20095 Milan, Italy; luca.persani@unimi.it (L.P.); c.colombo@auxologico.it (C.C.); 3Department of Medical Biotechnology and Translational Medicine, University of Milan, 20133 Milan, Italy; 4Department of Pathophysiology and Transplantation, University of Milan, 20122 Milan, Italy

**Keywords:** papillary thyroid cancer, clonality, tumor heterogeneity

## Abstract

Despite its potential clinical impact, intra-tumor genetic heterogeneity (ITH) has been scantly investigated in papillary thyroid cancer (PTC). We studied ITH in PTC by combining, for the first time, data derived from the evaluation of the normalized allelic frequencies (NAF) of the mutation/s, using a customized MassARRAY panel, and those obtained by the HUMARA clonality assay. Among tumors with a single mutation, 80% of cases with NAF 50 ± 5% were clonal, consistent with the presence of a single mutated clone, while 20% of cases showed a polyclonal pattern, suggesting the presence of the same mutation in two or more clones. Differently, all cases with NAF < 45% were polyclonal. Among tumors with double mutation, cases with both mutations showing NAF 50 ± 5% were monoclonal, consistent with the presence of a single clone harboring both mutations. On the other hand, all cases with double mutation at NAF < 45% were polyclonal, indicating the presence of two clones with different mutations. Finally, no significant differences in the clinico-pathological characteristics were found between monoclonal and polyclonal tumors. In conclusion, the present study adds insights into the concept of ITH in PTC, which warrants attention because the occurrence of this phenomenon is likely to affect the response to targeted drugs.

## 1. Introduction

Intra-tumor genetic heterogeneity (ITH) refers to the coexistence of genetically different subclonal populations within the same tumor. ITH could have a significant impact on prognosis and on the response to targeted therapy, especially in the era of personalized medicine [1]. Indeed, the clinical and therapeutic decisions are commonly based on the mutation/s found in the primary tumor that might not be representative of the genetic pattern within a tumor as a whole, because it could evolve during tumor progression, mainly due to the selective pressure of treatment [2]. Despite its potential clinical relevance, ITH has been scantly investigated in papillary thyroid cancer (PTC) [3], probably because the overall density of somatic mutations is lower than in other cancers [1]; however, a recent study revealed a median of 40% of mutational ITH in PTC [4].

In recent years, thanks to advances in next-generation sequencing (NGS) technology, it has become evident that multiple alterations can be concomitantly present in PTC. In particular, the coexistence of different point mutations or of point mutations and fusions, e.g., *BRAF*^V600E^ and *RET/PTC* or *BRAF*^V600E^ and *TERT* promoter were found to occur in up to 20% of PTCs [5,6,7,8,9,10]. Interestingly, PTCs with dual mutations were associated with older age at diagnosis and a worst disease outcome, suggesting that these tumors undergo a positive selection and are more aggressive [7,8,9,10,11]. The origin of concomitant mutations in PTC, i.e., from single or multiple clones, is an unsolved issue. Of note, most of the clonality studies reported on PTC were focused on the clonal relations between foci in multifocal PTCs [12], and the few available studies on isolated PTC cases mainly indicated a monoclonal pattern [13,14]. Consistently, the Cancer Genome Atlas Consortium (TCGA) showed that the majority of all driver mutations have a calculated cancer cell fraction close to 1.0, suggesting their presence in all tumor cells [15].

In contrast, other studies showed that *BRAF*^V600E^ and *TERT* promoter mutations can be heterogeneously distributed in PTC, indicating their possible subclonal or even oligoclonal occurrence [9,16,17,18]. The correlation between the clonal status of PTC with the clinico-prognostic feature of the patients is controversial. In particular, some studies reported that subclonal *BRAF* mutations associate with smaller tumors [16,17,18], less frequent extrathyroidal extension [17,18] and lower recurrence rate [16], while other reports did not find significant association with disease progression [19]. On the contrary, the high burden of subclonal mutations has been associated with distant metastases and increased risk of relapse or death [4].

We have previously characterized a large series of PTCs by a custom MassARRAY panel (PTC-MA), finding at least one molecular alteration in 71% of cases. Interestingly, in 19% of cases two or more mutations were detected, and a minority of cases were expected to have subclonal mutations, consistent with ITH [9].

In the present study we aimed to investigate ITH in PTC by combining, for the first time, data derived from the evaluation of the allelic frequencies of the mutation/s, using a customized MassARRAY panel, and those obtained by the HUMARA clonality assay. Moreover, we correlated the clonality status of the tumors with the clinico-pathological characteristics.

## 2. Methods

### 2.1. Patients

This is a retrospective cohort study with institutional review board approval and informed patient consent for the use of thyroid tumor tissues and collection of clinico-pathological information. Criteria for inclusion of PTC samples were: (a) availability of ipsilateral normal tissue; (b) data on the neoplastic cell content; (c) presence of a point mutation; (d) availability of the percentage of mutated alleles. Upon these criteria, from PTC samples either included in our previously published PTC series [9] or more recently studied, we selected 88 female cases with a tumor purity > 70% in order to avoid or maximally reduce the contamination from normal thyroid tissue. As shown in Figure 1, 35 cases were excluded due to the absence of known point mutations found in the tumor sample, and the remaining 53 cases were submitted to further analyses. All the patients included were followed in a single tertiary care endocrine center during the period 1990–2020, and were diagnosed and treated according to the recent guidelines for the management of thyroid cancer [20,21]. Tumors were classified and staged according to the thyroid malignancy World Health Organization classification and the 8th edition of TNM staging [22]. Clinico-pathological features at diagnosis and the final disease outcome, after a mean follow-up of 56 months (range 6–225), were available for all included patients.

### 2.2. DNA Extraction

DNA extraction was performed on formalin fixed paraffin embedded (FFPE) tissues. Since a high fraction of non-tumor cells in the sample could lead to the detection of a “false” polyclonality, only cases with a percent of tumor cells > 70% were selected. Hematoxylin and eosin sections next to the sample used for genetic analyses were evaluated by two pathologists to define tumor purity in each sample [9]. This procedure ensures the selection of the core of the tumor and excludes the possibility of multifoci sampling, similar to microdissection. The percentage of neoplastic cells on the whole of the cell content, thus including surrounding stromal and immune/inflammatory cells, was defined. The mean percentage of cancer cells was calculated in each sample by looking at 100 cells in 4 fields at 40× magnification. For each field, the number of tumor cells per 100 cells were counted and a mean of the results obtained in the 4 fields was obtained.

Genomic DNA was extracted from formalin fixed paraffin embedded (FFPE) tissues, after xylene deparaffination using a commercial kit (Puregene^®^ Core Kit A, Qiagen, Germantown, MD, USA), following the manufacturer’s instructions.

### 2.3. Genotyping

DNA was analyzed using the custom PTC-MA assay, based on MALDI-TOF Mass spectrometry (Agena Bioscience, San Diego, CA, USA), previously set up by our group [23]. In particular, PCR amplification and extension primers were designed to interrogate 13-point mutations in 8 genes (*BRAF*, *H-RAS*, *N-RAS*, *K-RAS*, *PIK3CA*, *TERT*, *EIF1AX*, *AKT*), using the Sequenom MassARRAY Assay Design 3.1 Software (Agena Bioscience, San Diego, CA, USA). This technology requires minimum amounts of DNA and works on short DNA sequences, as obtained from FFPE tissues [23].

The allelic frequency calculated by the software was normalized for the cancer cell content (normalized allelic frequency, NAF). Because the sensitivity of the MassARRAY technology is 5% [23], samples with heterozygous point mutations at NAF 50 ± 5% were expected to be clonal, while those at NAF < 45% were expected to be polyclonal.

Two single nucleotide polymorphisms (SNPs), rs7801086 and rs10232557, located in intron 13 and 16 of the *BRAF* gene, respectively, were investigated by PCR and Sanger sequencing on a 3500 Genetic Analyzer (Applied Biosystem, Foster City, CA, USA), using the following primers: rs7801086F: TTGGGAAGAATGAGGACGTTT, rs7801086R: TCTTACTCTATCACCCAGGCTG, rs10232557F: GAGAGGTAGATTAACAGCCTT, rs10232557R: TCCTGGGATCAAGTGATCCT.

### 2.4. HUMARA Assay

The assessment of clonality was performed using the HUMARA (human androgen receptor gene, Xq13) assay, following the protocol described by Allen et al. (1992) [24], with a few modifications. A high polymorphic CAG microsatellite repeat is located in the first exon of the HUMARA gene, close to restriction sites of HpaII, which is a methylation-sensitive endonuclease. In females, the X-chromosome inactivation is associated with the methylation of those restriction sites. Digestion with HpaII, followed by PCR amplification of the microsatellite repeat with primers flanking the restriction sites, results in the amplification only of the transcriptional inactive (methylated) allele. In informative females (heterozygous for CAG repeat numbers), the presence of 2 PCR products of different sizes, derived from the amplification of both allelic CAG repeats, indicates a polyclonal cell population, while a monoclonal population will result in a single PCR product.

We incubated 500 ng of DNA with 50 U of the enzyme Hpa II (New England Biolabs, Ipswich, MA, USA) at 37 °C for 16 h. For each sample we included a negative control containing only the DNA, but no enzyme. Moreover, for each experiment we included a hemizygous (male thyroid tissue) control sample. Paired digested and undigested samples were amplified using a GoTaq^®^ Hot Start Polymerase kit (Promega, Madison, WI, USA) with the following primers: forward: [5′-FAM-labelled] ACCGAGGAGCTTTCCAGAAT; reverse: TGGGGAGAACCATCCTCA. Thermal cycling conditions were set as the following: the initial denaturation step at 95 °C for 5 min, followed by 40 cycles of denaturation at 95 °C, annealing at 55 °C and extension at 72 °C all for 1 min, with a final extension at 72 °C for 7 min. PCR products were size separated on an automatic sequencer (3500 Genetic Analyzer, Applied Biosystems, Foster City, CA, USA) and results were analyzed using GeneMapper5 software (Applied Biosystems, Foster City, CA, USA). The allelic ratio (AR) was calculated by dividing the peak area of the shorter allele by that of the longer allele. To correct for a possible preferential amplification of one allele, we calculated the corrected AR (CR) by dividing the AR of each HpaII digested sample by the AR of the corresponding undigested control. If the CR was ≤0.33 or ≥3, the sample was defined as clonal.

Normal thyroid epithelium is known to be organized into large stem cell-derived monoclonal patches [25]; therefore, monoclonality in neoplastic lesions might simply correspond to the clonal composition of the normal epithelium. To exclude the possible occurrence of a non-random (skewed) X-inactivation in the tumor tissues analyzed, leading to the non-reliability of a monoclonal result, we also performed the HUMARA assay in the corresponding ipsilateral normal tissues.

### 2.5. Statistical Analyses

Relations between discrete variables were evaluated by means of the χ^2^ test or *t*-test, as appropriate. Clinico-pathological and molecular features were evaluated by a univariate analysis. Statistical significance was defined as *p* <  0.05. All statistical analyses were performed using MedCalc Analyses using the Version 18.11.3 of the MedCalc Software (B-8400, Ostend, Belgium).

## 3. Results

### 3.1. Mutational Analysis

Among the 53 cases selected based on the female gender, the neoplastic cell content > 70%, and the presence of at least 1-point mutation, a single *BRAF^V600E^* mutation was found in 23 cases, a single mutation in codon 61 of *H-* or *N-RAS* in five PTCs and a single *TERT c.-124C>T* mutation in two PTCs, whereas 12 cases harbored both *BRAF*^V600E^ and *TERT* c.-124C>T mutations (Table 1). Among the 30 tumors with a single mutation, 16 had normalized allelic frequency (NAF) of 50 ± 5%, while the remaining had NAF < 45% (range 13–44). Among the 12 cases with a double mutation, five had NAF 50 ± 5% for both mutations, one (#36) had NAF 50% and 80%, respectively, and six had NAF < 45% for one or both mutations.

Based on the NAFs, the cases with a single or double mutation of 50 ± 5% were expected to be monoclonal, either with one or two clonal mutations. On the other hand, those with a single or double mutation at NAF < 45% were classified as subclonal, with the exception of two cases (#41 and #42) harboring one clonal mutation and one subclonal mutation, each.

### 3.2. HUMARA Clonality Assay

To further investigate the clonal status of the mutated tumors, we performed the HUMARA clonality assay. Of the 53 tissues analyzed, 42 (79%) were informative for the assay, while 11 were homozygous for CAG repeat numbers and thus non-informative and excluded from further analyses (Figure 1). The HUMARA assay showed that 21 tumors were polyclonal and 21 monoclonal, though five monoclonal cases were excluded because the ipsilateral normal thyroid tissue showed a skewed pattern of X-inactivation, suggesting the presence of an embryonic patch size (Table 1, Figure 1).

Cases with a single mutation (*n* = 26) resulted either poly- or monoclonal (Table 1, Figure 1). In particular, the majority of cases (12/15, 80%) with NAF 50 ± 5% were clonal, consistent with the existence of a single mutated clone in the tumor (Figure 2, panel A). Nevertheless, three *BRAF* mutated cases expected to be clonal had a HUMARA polyclonal pattern (#14, #15, #16) (Table 1, Figure 1). To further confirm the existence of multiple BRAF mutated clones in these three tumors, we examined two SNPs (rs7801086 and rs rs10232557) in intron 13 and 16 of the *BRAF* gene, respectively (minor allele frequency: 35%, 1000 Genomes Project), with the aim to analyze the segregation of the two SNPs with *BRAF*^V600E^ mutation after subcloning, as reported [26]. Unfortunately, the three samples were not informative for the two *BRAF* SNPs examined and the presence of multiple *BRAF*^V600E^ mutated clones could not be verified.

On the other hand, all cases harboring a single mutation with NAF < 45% (*n* = 11) were polyclonal. In these cases, we hypothesized the presence of different clones, one harboring the detected mutation and one or more with unknown genetic drivers (Table 1, Figure 1 and Figure 2, panel B).

Among tumors with double mutation, four cases with both mutations showing NAF 50 ± 5% were monoclonal (Table 1, Figure 1 and Figure 2, panel C). Differently, one case (#36) harboring *BRAF*^V600E^ and *TERT* c.-124C>T mutations, with NAF of 50 and 80%, respectively, was polyclonal.

Interestingly, all cases harboring a double mutation with at least a variation showing NAF < 45% (*n* = 6) were polyclonal. This finding may indicate the presence of subclones with two different mutations or, alternatively, the presence of a subclone with both mutations and of a subclone with unknown mutations (Table 1, Figure 1 and Figure 2, panel D).

### 3.3. Correlations between Clonal Status and Clinico-Pathological Features

We correlated the clonal status of the tumors with their genetic background and several clinico-pathological characteristics of the patients (Table 2). No significant differences were found between monoclonal and polyclonal tumors, with the exception of the gross extrathyroidal extension (ETE) and lymph node metastases (N1), which were more frequent, though not reaching the statistical significance, in polyclonal tumors (57 vs. 25% and 69 vs. 30%, respectively). Radioiodine therapy (RAI) was performed more frequently in patients with polyclonal tumors (76% vs. 38%, *p* = 0.019). The final disease outcome did not differ between the two groups of tumors (disease persistence of 19% in monoclonal vs. 24% in polyclonal PTCs), even upon separation of patients treated or not with RAI.

## 4. Discussion

Although the majority of driver mutations are found in tumors as a clonal event, a fraction of them occur subclonally. It has been estimated that some degree of ITH is likely present in all solid tumors [28]. ITH may be considered an indicator of a tumor’s potential to evolve under selective pressure. Consistently, pan-cancer analyses have demonstrated that ITH has an independent prognostic value, being associated with higher stage, worst outcome and poorer patient survival [29,30]. PTC is known to harbor a relatively low burden of somatic mutations compared to other solid tumors, and this may represent the biological explanation of its indolent clinical behavior. Nevertheless, more recent methods of genetic analysis, i.e., single-cell sequencing, and the multi-region NGS approach, made possible a deeper and more extensive analysis of the mutational status of tumor samples, providing data in favor of ITH in PTC [1]. The pattern of PTC clonality may have potential clinical implications on individualizing the prognosis. Indeed, a high burden of subclonal mutations has been associated with increased risk of relapse [4], and in our series polyclonal cases had a higher prevalence of extrathyroidal invasion and lymph nodal metastases. It is also well known that the presence of ITH is the cause of tumor relapse after successful therapy [31]. Indeed, a fraction of the cells within the heterogeneous population are predicted to be drug resistant, able to survive treatment and expand over time. Another cause of “escape” from the initial response to a therapeutic compound, is the temporal heterogeneity, due to the progressive increase of the mutational burden during the de-differentiation process [32,33]. Interestingly, recent evidence indicated that the occurrence of secondary RAS mutations is involved in resistance to BRAF inhibitors in thyroid cancer [34,35].

In the present work, we demonstrated the presence of ITH in PTC by combining data obtained by evaluating the normalized mutated allelic frequencies with those derived from the HUMARA assay. In particular, 16/30 tumors with a single mutation and 5/12 tumors with double mutation had NAF 50 ± 5%, consistent with the presence of the heterozygous mutation/mutations in all the neoplastic cells. In these cases, the HUMARA assay confirmed monoclonality in all but three cases with a single mutation, which were polyclonal. Since contamination of non-tumor cells was minimized in our samples, the three polyclonal cases with a single *BRAF*^V600E^ mutation (#14, #15, #16) are predicted to harbor the same mutation in two or more clones. Although it could not be verified after subcloning due to the lack of informative *BRAF* SNPs, this finding is consistent with what is already described in melanoma. Indeed, single cells genotyping showed that *BRAF* activating mutations may coexist in different cells of the same melanoma [26]. At variance, case #36, harboring both *BRAF* and *TERT* mutations with NAF 50% and 80%, respectively, was polyclonal, and we hypothesized the presence of two different clones, one with clonal *BRAF*^V600E^ mutation and one with *TERT* c.-124C>T mutation at high NAF. The finding of NAF > 55% for the *TERT* mutation likely indicates the loss of heterozygosity (LOH) of the wild type allele, as previously shown [36].

On the other hand, 11 cases with one mutation had NAF < 45%, consistent with the presence of different clones, as further confirmed by the HUMARA assay which showed polyclonality in all these cases.

Finally, the six remaining cases with coexisting alterations had NAF < 45% in one or both mutated genes, thus suggesting the existence of distinct clones harboring different mutations, consistent with the presence of ITH. In all these cases, the HUMARA clonality assay confirmed the presence of multiple clones. In the two cases with NAF 50 ± 5% for one mutation and NAF < 45% for the other (#41, #42) we hypothesized the presence of a clone with only one mutation and a clone with both mutations.

No correlation was found about the clonal status and the clinical and outcome features of the patients. Nevertheless, polyclonal cases had a higher, though not statistically significant, prevalence of ETE and N1, and were thus more frequently treated with RAI. However, the disease outcome was not significantly different between the two groups of PTCs, also likely due to the known good clinical outcome of tumors and the limited number of cases analyzed.

This study has some limitations. The HUMARA assay is considered an accurate and simple method to assess clonality in samples from female subjects [24], and has been used and validated in thyroid [12] and other cancers [37]. Nevertheless, it should be highlighted that, in the HUMARA analysis, polyclonality may result from a high fraction of non-tumor cells (stroma, lymphocytes). To overcome this potential weakness, we included only cases with a high tumor purity threshold. On the other hand, though we excluded the presence of patch size, the existence of different clones with the same pattern of X-inactivation should not be excluded in monoclonal cases.

Furthermore, the mutational analysis was not based on an NGS approach. Nevertheless, our target panel covered 80% of the driven mutations reported in PTC [15] and was able to provide the level of heterogeneity of the tumor tested [23]. However, the presence of passenger or very rare mutations, and consequently information on ITH, could be missed with our technology in a minority of samples. Of note, frequencies of mutated alleles were always normalized for the percentage of tumor cells. Therefore, we are confident that subclonal mutations were not the result of the presence of non-tumor cells in the sample analyzed.

Another drawback relates to the recent finding of either clonal or subclonal driver mutations in different regions of PTCs, which highlights the importance of multi-region sampling [4]. Our analysis was performed on a single region of the tumors; therefore, it is possible that some information about ITH was missed.

Finally, as a consequence of our strict selection criteria, the sample size was likely too small to find a significant correlation with clinical and prognostic data. Thus, a very large series would be needed to get more insights into this aspect, being PTC poorly aggressive and very effectively curable.

In conclusion, we have given major insights, obtained by the combination of two different tools, into the concept of ITH in PTC. Half of the tumors analyzed were found to be polyclonal, with two or more foci harboring the same or different mutations. The heterogeneity found in some tumors warrants attention, since the occurrence of this phenomenon is likely to affect the response to targeted drugs [34,35].

## Figures and Tables

**Figure 1 jcm-10-02645-f001:**
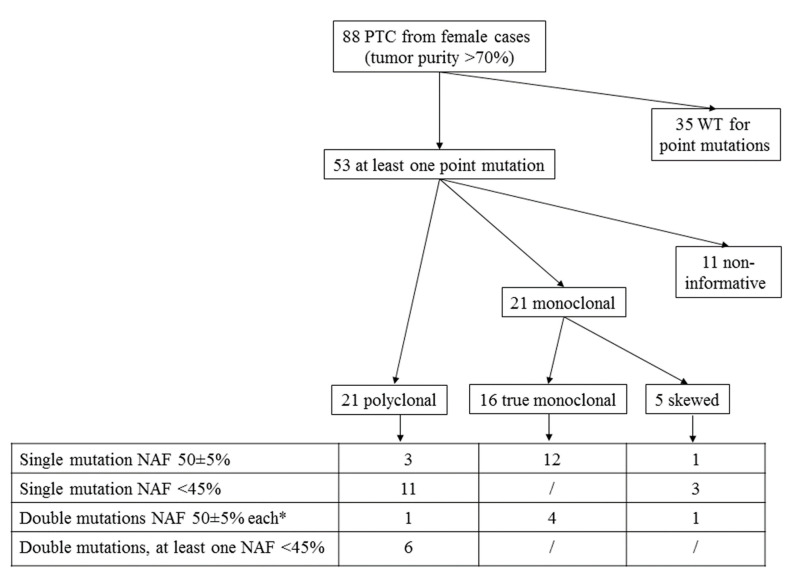
Case selection and results of HUMARA clonality assay. NAF: normalized allelic frequency. * One case had one mutation with NAF 50% and the other with NAF 80% likely indicating the loss of heterozygosity (LOH) of the wild type allele.

**Figure 2 jcm-10-02645-f002:**
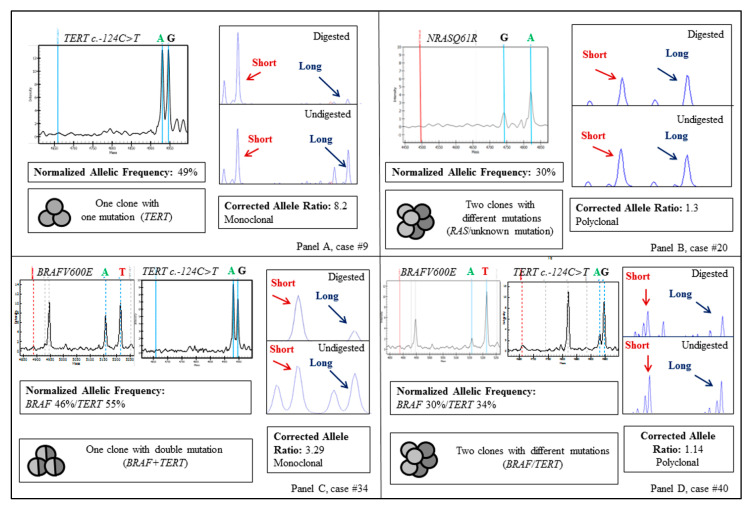
MassARRAY spectra and HUMARA assay electropherograms obtained in: one monoclonal case with single clonal mutation (**panel A**), one polyclonal case with single subclonal mutation (**panel B**), one monoclonal case with double clonal mutation (**panel C**) and one polyclonal case with double subclonal mutation (**panel D**). Panel D represents the most plausible scenario, considering that thyroid cancer is predicted to have a low mutational burden [27]. Nevertheless, the presence of two clones, one with double *BRAF* and *TERT* mutation and one with unknown mutation is also possible. TERT c.-124C>T mutation was detected by MassARRAY using a reverse primer.

**Table 1 jcm-10-02645-t001:** Normalized allelic frequencies of mutations detected by the MassARRAY panel and clonality status identified by the HUMARA assay in 42 PTCs. Tumors 1–16: single mutation with normalized allelic frequency (NAF) 50 ± 5%, tumors 17–30: single mutation with NAF < 45%, tumors 31–36: double mutations with NAF 50 ± 5%, tumors 37–42: double mutations, with at least one with NAF < 45%. Cases 13, 17, 18, 19, 35: non-random (skewed) X-inactivation detected in the contralateral normal tissues. #, Case.

#	Normalized Allelic Frequency			Clonality (Corrected Ratio)
	*BRAF* V600E	*TERT* c.-124C>T	*N-RAS* Q61K/R/ *H-RAS* Q61R	
1			50	monoclonal (0.3)
2	50			monoclonal (0.08)
3	50			monoclonal (0.13)
4	55			monoclonal (0.24)
5	45			monoclonal (16.14)
6	48			monoclonal (5)
7			45	monoclonal (0.29)
8	49			monoclonal (9.13)
9		49		monoclonal (8.16)
10			52	monoclonal (3.1)
11	46			monoclonal (4.31)
12	53			monoclonal (0.08)
13		47		monoclonal (4.2), normal tissue skewed (3.5)
14	47			polyclonal (0.68)
15	53			polyclonal (0.6)
16	48			polyclonal (1.27)
17			15	monoclonal (12.3), normal tissue skewed (3.2)
18	13			monoclonal (5.18), normal tissue skewed (18.8)
19	31			monoclonal (8.33), normal tissue skewed (3)
20			30	polyclonal (1.33)
21	44			polyclonal (0.73)
22	43			polyclonal (1.15)
23	12			polyclonal (0.8)
24	17			polyclonal (1.6)
25	36			polyclonal (0.91)
26	36			polyclonal (0.62)
27	35			polyclonal (1.66)
28	29			polyclonal (2.73)
29	40			polyclonal (1.18)
30	37			polyclonal (0.91)
31	49	50		monoclonal (8.11)
32	53	50		monoclonal (0.16)
33	49	49		monoclonal (0.16)
34	46	55		monoclonal (3.29)
35	45	53		monoclonal (18.8), normal tissue skewed (66.7)
36	50	80		polyclonal (2.38)
37	14	40		polyclonal (1.32)
38	23	39		polyclonal (0.76)
39	35	20		polyclonal (1.17)
40	30	34		polyclonal (1.14)
41	53	21		polyclonal (0.78)
42	15	50		polyclonal (1.34)

**Table 2 jcm-10-02645-t002:** Comparison of molecular and clinico-pathological features of the 16 monoclonal and the 21 polyclonal PTCs analyzed. PTC, papillary thyroid carcinoma; TNM, tumor, node, metastasis; AJCC, American Joint Committee on Cancer. * excluding the NX category.

Features (%)	Monoclonal PTCs (*n* = 16)	Polyclonal PTCs (*n* = 21)	*p* Value
*BRAF* ^V600E^	12/4 (75/25%)	20/1 (95/5%)	0.074
H-/N-RAS codon 61	3/13 (19/81%)	1/20 (5/95%)	0.174
*TERT* c.-124C>T	5/11 (31/69%)	7/14 (33/67%)	0.893
*BRAF*^V600E^ + *TERT* c.-124C>T	4/12 (25/75%)	4/17 (19/81%)	0.663
Median age at diagnosis, years (range)	45.5 (15–77)	56 (24–87)	0.747
Pre-surgical diagnosis, yes/indeterminate/no	14/1/1 (88/6/6)	21/0/0 (100/0/0%)	0.249
Size, mm (mean)	25.4 (8–55)	23.7 (8–44)	0.268
Extrathyroidal extension, yes/no	4/12 (25/75)	12/9 (57/43%)	0.053
Multifocality, yes/no	7/9 (44/56)	8/13 (38/62)	0.732
Histological variants of PTC, classical/follicular/other	12/3/1 (75/19/6)	18/0/3 (86/0/14%)	0.099
T1/T2/T3/T4	9/2/4/1 (56/12/25/7%)	9/6/3/3 (43/29/14/14%)	0.472
N1/N0/NX	3/7/6 (18/44/38%) 3/7 * (30/70%)	11/5/5 (52/24/24%) 11/5 * (69/31%)	0.110 0.053
M1/M0	1/15 (6/94%)	0/21 (0/100%)	0.251
AJCC Stage, I/II/III/IV	14/0/2/0 (88/0/12/0%)	14/4/2/1 (67/19/10/4%)	0.220
Radioiodine Ablation, yes/no	6/10 (38/42%)	16/5 (76/24%)	0.019
Disease outcome, persistence/remission	3/13 (19/81%)	5/16 (24/76%)	0.714
Disease-specific mortality, yes/no	1/15 (6/94%)	0/21 (0/100%)	0.251
Follow-up, months (mean, range)	62 (6–217)	54 (6–225)	0.145

## Data Availability

The data presented in this study are available on request from the corresponding author. The data are not publicly available for ethical and privacy reasons.

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
