# Peer review of "Combined Mutational and Clonality Analyses Support the Existence of Intra-Tumor Heterogeneity in Papillary Thyroid Cancer"

_jcm, 2021, doi:10.3390/jcm10122645_

Round 1

Reviewer 1 Report

I think this is the nice study although small.

I think we need to revise the conclusion and impact as currently there is no data in thyroid cancer and nor was any shown in your paper that patients with hetrogenity had different outcomes or response to targeted therapy.

You may want to expand on its impact in other tumors and how outcomes were influenced.

Author Response

I think this is the nice study although small.

We thank the Reviewer for her/his appreciation of our study

I think we need to revise the conclusion and impact as currently there is no data in thyroid cancer and nor was any shown in your paper that patients with heterogenity had different outcomes or response to targeted therapy. You may want to expand on its impact in other tumors and how outcomes were influenced.

We thank the Reviewer for her/his suggestion. Actually, heterogeneity likely has a major impact on tumor outcome and response to treatment. We expanded this topic in the Discussion (see page 14, lines 270-276, Reffs: 31-34)

Reviewer 2 Report

This paper reports an analysis of papillary thyroid carcinomas to determine whether they have intratumoral heterogeneity.  The authors evaluated both the normalized allelic frequencies (NAF) of mutations and the results of the HUMARA clonality assay. They report that

  • Among 29 cases with a single mutation, 80% with NAF 50±5% were clonal, 20% were polyclonal suggesting multiple clones; all cases with NAF <45% were polyclonal
  • Among cases with two mutations, cases with both mutations showing NAF 50±5% were monoclonal, whereas those with NAF <45% were polyclonal.

These variations had no clinical significance.

The data are interesting but are not surprising or unexpected. Firstly, it is well known that second mutations often arise in subclones of tumors.  In addition, thyroid carcinomas are frequently multifocal ab initio, so the likelihood of multiple lesions growing into a single clinically detectable tumor is not unlikely.

The authors do not offer any evidence of a careful approach to the characterization of the tumors they examined.  They define as the only requirement 70% tumor cell content.  However, they did not do microdissection to define specific areas of a tumor for analysis.  This may explain the collection of multifocal distinct primary tumors in a single piece of frozen tissue.

Most importantly, the presence of a driver mutation (eg BRAF, RAS) along with a secondary event (such as TERT promoter mutation) is well recognized in almost all studies to be a feature of subclonal progression. 

The lack of clinical significance of the results dampens enthusiasm for the value of this study.

Author Response

Q1. The authors do not offer any evidence of a careful approach to the characterization of the tumors they examined.  They define as the only requirement 70% tumor cell content.  However, they did not do microdissection to define specific areas of a tumor for analysis.  This may explain the collection of multifocal distinct primary tumors in a single piece of frozen tissue.

R1. As reported in the methods, DNA extraction was performed on formalin fixed paraffin embedded (FFPE) tissues. Hematoxylin & eosin sections next to the sample used for genetic analyses were evaluated by two pathologists to define tumor purity in each sample. This procedure ensures the selection of the core of the tumor and excludes the possibility of multifoci sampling, similarly to microdissection. This has been added to the methods section (see page 5, line 116 and page 6, lines 120-121).

Q2. Most importantly, the presence of a driver mutation (eg BRAF, RAS) along with a secondary event (such as TERT promoter mutation) is well recognized in almost all studies to be a feature of subclonal progression. 

R2. Although it is well known that ITH develops during the process of dedifferentiation of thyroid cancer, the presence of ITH in early stage is debated [Fugazzola et al., Cancer 2020]. In particular, TERT promoter mutation was reported to be frequently sub-clonal in PTC, but the subclonal occurrence of BRAFV600E is controversial.  Our data indicated that the majority of the main mutations documented in PTC may be subclonal. The ITH of BRAF and RAS mutants calls into question their role of driver mutations and initiators of thyroid carcinogenesis. Moreover, in half of cases with double mutation we found, for the first time, the co-existence of TERT and BRAF mutations in the same clone. Understanding the clonal distribution of mutations in primary tumors may help to predict the genetic asset of subsequent metastases, and therefore to select the best treatment.

Q3. The lack of clinical significance of the results dampens enthusiasm for the value of this study.

R3. We do not completely agree with this comment. Data on clonality in thyroid tumors are scanty and controversial and we believe to have added new insights into this topic with our study. Similarly, almost no data exist on the impact of clonality on the outcome of these tumors. We know that thyroid tumors have an excellent prognosis in the vast majority of cases and this is likely based on factors not involving the clonal pattern, but including the low aggressiveness of the driver mutations, the well standardized methods of diagnosis, the availability or highly effective treatments such as thyroidectomy and radioiodine. On the other hand, in the revised version of the manuscript we report that, after excluding NX cases, polyclonal tumors are more frequently N1 than monoclonal cases (69% versus 30%, P = 0.053). Thus, though not statistically significant likely due to the limited number of cases, polyclonal tumors have a higher prevalence of both gross extrathyroidal extension and lymph nodal metastases (see revised Table 2, results page 12, lines 246-247, and discussion page 14, lines 269-270 and page 15, lines 300-304).

Reviewer 3 Report

The authors are to be congratulated for this very interesting work on an understudied topic. I do have some concerns which I hope the authors will be able to address:

  1. The targeted mutation analysis is a limitation of the study. In particular, the use of the MassARRAY panel rather than NGS is unexpected. Is there a benefit to the MassARRAY approach here instead of NGS, other than cost and availability of the technology?
  2. A whole exome or other less targeted genomic assay might be of interest for assessing clonality in these samples. So-called "passenger" mutations might serve as markers of clones, even if the "driver" mutations are of more clinical and biological significance.
  3. It was somewhat surprising that there was no relationship between clinicopathological factors and clonality, given that there was a difference in the proportion of patients who received RAI based on clonality. Presumably RAI was given to the patients who got it for some clinicopathological reason! I did notice that there may be an effect in terms of lymph node metastasis. In Table 2, Nx is treated as its own category for the purposes of statistical analysis. In fact I believe it would be more appropriate to exclude Nx patients from that analysis and study just N1 vs N0 (since Nx patients are essentially uninformative with regard to lymph node status). This would reveal that polyclonal cases had much higher rates of lymph node positivity, a finding of potential clinical significance (though it may not be statistically significant in light of the modest cohort size).
  4. For the cases with multiple mutations that are subclonal, the diagram in Figure 2D is too simple (and the explanation of those cases in the text and caption). We do not know that there are 2 sub-populations with one mutation each - maybe there is a sub-clone with both mutations, and the other sub-clone has neither.
  5. The use of the HUMARA assay to assess clonality is interesting. The approach seems predicated somewhat on methylation of one allele as would be expected in most healthy/normal cells containing two X chromosomes. However, in cancer cells there may be significant alterations in methylation patterns which could significantly change the interpretation of the results of this assay. If there have been previous studies that have demonstrated/validated this assay in cancer, they should be added as references.
  6. Is there any role of the SNPs mentioned in the BRAF introns (lines 135-140)? It seems the cases where it was performed were uninformative and the data from that experiment was not included in the manuscript. Should the description of that experiment be removed?
  7. Within the abstract and introduction, it would be good to clarify that the "mutations" being studied are derived from a limited/targeted panel of known driver mutations. This is explained in the methods but not as clear from the start how exactly the mutations were profiled.

Author Response

The authors are to be congratulated for this very interesting work on an understudied topic.

We thank the Reviewer for her/his appreciation of our study

I do have some concerns which I hope the authors will be able to address:

Q1. The targeted mutation analysis is a limitation of the study. In particular, the use of the MassARRAY panel rather than NGS is unexpected. Is there a benefit to the MassARRAY approach here instead of NGS, other than cost and availability of the technology?

Q2. A whole exome or other less targeted genomic assay might be of interest for assessing clonality in these samples. So-called "passenger" mutations might serve as markers of clones, even if the "driver" mutations are of more clinical and biological significance.

R1 and R2. We agree with these comments. Nevertheless, with our customized panel we are able to detect all the most frequent and known genetic alterations of thyroid cancer without any bioinformatic analysis. Moreover, this technology requires minimum amounts of DNA and works on short DNA sequences, as obtained from FFPE tissues [Pesenti et al., Endocrine 2018]. It is certainly possible that passenger or very rare mutations could be missed with our technology in a minority of samples.  We highlighted this as a drawback of the study in our conclusion (see page 15, lines 314-315). 

Q3. It was somewhat surprising that there was no relationship between clinicopathological factors and clonality, given that there was a difference in the proportion of patients who received RAI based on clonality. Presumably RAI was given to the patients who got it for some clinicopathological reason! I did notice that there may be an effect in terms of lymph node metastasis. In Table 2, Nx is treated as its own category for the purposes of statistical analysis. In fact, I believe it would be more appropriate to exclude Nx patients from that analysis and study just N1 vs N0 (since Nx patients are essentially uninformative with regard to lymph node status). This would reveal that polyclonal cases had much higher rates of lymph node positivity, a finding of potential clinical significance (though it may not be statistically significant in light of the modest cohort size).

R3. We thank the Reviewer for this comment. Patients underwent RAI according to criteria defined by International guidelines in a uniform approach in all patients [Haugen et al., Thyroid 2016 and Pacini et al., J. Endocrinol. Invest. 2018]. Interestingly if we exclude NX cases, as suggested by the Reviewer, polyclonal PTCs result more frequently N1 than monoclonal cases (69% versus 30%, P = 0.053). Thus, though not statistically significant, polyclonal cases have a higher prevalence of both gross extrathyroidal extension and lymph nodal metastases. Therefore, we modified accordingly the Table 2 and the text (see revised Table 2, results page 12, lines 246-247, and discussion page 14, lines 269-270 and page 15, lines 300-304).

Q4. For the cases with multiple mutations that are subclonal, the diagram in Figure 2D is too simple (and the explanation of those cases in the text and caption). We do not know that there are 2 sub-populations with one mutation each - maybe there is a sub-clone with both mutations, and the other sub-clone has neither.

R4. We agree with this comment and we modified caption of Figure 2D and text (pag 11, lines 226-229 and pag 12, lines 239-241) accordingly. For graphical clarity, in Figure 2D we represented the most plausible scenario, considering that thyroid cancer is predicted to have an average of two genetic drivers per tumor (Ref 37 added).

Q5. The use of the HUMARA assay to assess clonality is interesting. The approach seems predicated somewhat on methylation of one allele as would be expected in most healthy/normal cells containing two X chromosomes. However, in cancer cells there may be significant alterations in methylation patterns which could significantly change the interpretation of the results of this assay. If there have been previous studies that have demonstrated/validated this assay in cancer, they should be added as references.

R5. Thank you for this comment. Actually, this assay has been used and validated in thyroid and other cancers. This has been reported in the Discussion and Reference added (see page 15, lines 305-307 and Ref 36).

Q6. Is there any role of the SNPs mentioned in the BRAF introns (lines 135-140)? It seems the cases where it was performed were uninformative and the data from that experiment was not included in the manuscript. Should the description of that experiment be removed?

R6. We fully understand the comment of the Reviewer. Indeed, the two SNPs examined (rs7801086 and rs10232557) in intron 13 and 16 of the BRAF gene, respectively, were not informative in our cases, and the presence of multiple BRAFV600E mutated clones could not be thus verified. Nevertheless, we wish to maintain the description of the experiment and the result, even if inconclusive, not to leave to the reader open questions.

Q7. Within the abstract and introduction, it would be good to clarify that the "mutations" being studied are derived from a limited/targeted panel of known driver mutations. This is explained in the methods but not as clear from the start how exactly the mutations were profiled.

R7. According to the suggestion, the method to detect mutations has been added in the Abstract and Introduction sections (see page 2, line 31 and page 4, lines 88-89).

Round 2

Reviewer 2 Report

This paper has been resubmitted with response to the previous reviews.

The authors have made no significant changes to the paper. They have only offered the alternative explanation stating:

“Panel D represents the most plausible scenario, considering that thyroid cancer is predicted to have a low mutational burden [37]. Nevertheless, the presence of two clones, one with double BRAF and TERT mutation and one with unknown mutation is also possible. “

While it is true that thyroid cancer is predicted to have a low mutational burden compared with other malignancies, it is also well documented that there are a few clear driver mutations that are primary mutations and additional events that increase the aggressiveness occur in a clinically relevant proportion of tumors.  In the example shown, BRAFV600E is a well-known common driver, and TERT is a well-recognized secondary mutation that predicts more aggressive disease.  Therefore they are not correct to call their interpretation “the most plausible scenario”.

This study should have been performed with single cell analysis to prove the hypothesis.  As stated previously, the lack of clinical significance of the results dampens any enthusiasm for the value of this study that has not definitively proven its claim.

Reviewer 3 Report

I wish to thank the authors for their edits and accommodations. See below for my comments.

Q1/Q2 - the text edits are adequate to clarify the method used and reasons for it. It might be useful to add a sentence at line 134 pointing out the advantage of this technology for mutational analysis from FFPE samples with short DNA fragments - would clarify why the platform was used, and potentially encourage other investigators to make use of it for this application.

Q3 - I appreciate the edits to the table and text. I believe this strengthens the manuscript's clinical relevance, though unfortunately the conclusions are still somewhat limited in statistical significance due to the small size of the cohort.

Q4 - I think the edits to caption and text are useful. Perhaps stating that PTCs are expected to have "low" mutational burden might illustrate the point rather than specifying "two" at line 227 - specifying two driver mutations per tumor is somewhat confusing since the present manuscript identified only one driver in most studied tumors.

Q5 - Great. 

Q6 - To me including the experimental approach in methods, when none of the results were interpretable or contribute to the conclusions, seems to raise more questions than it answers. I would again suggest to remove that portion but will defer to the authors and editors.

Q7. Thanks - this adds some clarity to those sections.

Author Response

Q1/Q2 - the text edits are adequate to clarify the method used and reasons for it. It might be useful to add a sentence at line 134 pointing out the advantage of this technology for mutational analysis from FFPE samples with short DNA fragments - would clarify why the platform was used, and potentially encourage other investigators to make use of it for this application.

R1 and 2) Thank you for the comment. According to this suggestion, we added the advantage of our technology at page 6, lines 134-135.

Q3 - I appreciate the edits to the table and text. I believe this strengthens the manuscript's clinical relevance, though unfortunately the conclusions are still somewhat limited in statistical significance due to the small size of the cohort.

R3) We agree with the Reviewer, the lack of significance may be due to the limited number of cases analysed.

Q4 - I think the edits to caption and text are useful. Perhaps stating that PTCs are expected to have "low" mutational burden might illustrate the point rather than specifying "two" at line 227 - specifying two driver mutations per tumor is somewhat confusing since the present manuscript identified only one driver in most studied tumors.

R4) Thank you for the suggestion. We substituted "an average of two driver mutations per tumor" with "a low mutational burden " at page 11, line 229.

Q5 - Great.

R5) Thank you for your appreciation.

Q6 - To me including the experimental approach in methods, when none of the results were interpretable or contribute to the conclusions, seems to raise more questions than it answers. I would again suggest to remove that portion but will defer to the authors and editors.

R6) According with the Reviewer's  suggestion,  we removed this experimental approach from the Method section. We left this point in the Results and Discussion, specifying that we were not able to prove our hypothesis due to the lack of informative SNPs in these cases.

Q7. Thanks - this adds some clarity to those sections.

R7) Thank you for your appreciation.